# Identification of Phlorotannins in the Brown Algae, *Saccharina latissima* and *Ascophyllum nodosum* by Ultra-High-Performance Liquid Chromatography Coupled to High-Resolution Tandem Mass Spectrometry

**DOI:** 10.3390/molecules26010043

**Published:** 2020-12-23

**Authors:** Roya R. R. Sardari, Jens Prothmann, Olavur Gregersen, Charlotta Turner, Eva Nordberg Karlsson

**Affiliations:** 1Division of Biotechnology, Department of Chemistry, Lund University, P.O. Box 124, 221 00 Lund, Sweden; eva.nordberg_karlsson@biotek.lu.se; 2Centre for Analysis and Synthesis, Department of Chemistry, Lund University, P.O. Box 118, 221 00 Lund, Sweden; jens.prothmann@chem.lu.se (J.P.); Charlotta.Turner@chem.lu.se (C.T.); 3Ocean Rainforest Sp/F, 20 Mjólkargøta, FO-180 Kaldbak, Faroe Islands; olavur@oceanrainforest.com

**Keywords:** extraction, *Saccharina latissima*, *Ascophyllum nodosum*, brown algae, polyphenols, phlorotannins, high-resolution mass spectrometry

## Abstract

Phlorotannins are bioactive polyphenols in brown macroalgae that make these algae interesting as healthy food. Specific phlorotannins are, however, seldom identified, and extracts from different species are often only analysed for total phenolic content (TPC). In this study, our focus was to identify phlorotannin molecules from *Saccharina latissima* and *Ascophyllum nodosum* (a species rich in these compounds) using ultra-high-performance liquid chromatography coupled to high-resolution tandem mass spectrometry (UHPLC-HRMS^2^). Water and ethanol (30 and 80% *v*/*v*) were used at solid:liquid ratios, extraction times and temperatures, proposed to result in high TPC in extracts from other species. The *S. latissima* extracts, however, did not allow phlorotannin detection by either UHPLC-UV/Vis or UHPLC-HRMS^2^, despite a TPC response by the Folin–Ciocalteu assay, pinpointing a problem with interference by non-phenolic compounds. Purification by solid phase extraction (SPE) led to purer, more concentrated fractions and identification of four phlorotannin species in *A. nodosum* and one in *S. latissima* by UHPLC-HRMS^2^, using extracts in ethanol 80% *v*/*v* at a solid:liquid ratio of 1:10 for 20 h at 25 °C with an added 10 h at 65 °C incubation of remaining solids. The phlorotannin with the formula C_12_H_10_O_7_ (corresponding to bifuhalol) is the first identified in *S. latissima*.

## 1. Introduction

Polyphenols are one of the essential micronutrients in the human diet due to their health effects, antioxidant properties, and prevention of diseases associated with oxidative stress such as cancer, cardiovascular, and neurodegenerative diseases [1]. In plants and macroalgae, they are produced as secondary metabolites that exhibit protective functions against: (i) Environmental (abiotic) factors via their antioxidant properties by protection against reactive oxygen species produced as a consequence of high UV exposure and (ii) herbivores (biotic factors) by complexing proteins and decreasing digestive efficiency [1,2,3,4]. Hence, there is increasing interest in these compounds and their exploitation as bioactive molecules from natural resources.

Marine brown macroalgae (seaweeds) have been recognized as one of the main natural sources for biologically active polyphenols [5]. In brown seaweeds (Phaeophysa), the major types of polyphenols are phlorotannins which are oligomers and polymers of phloroglucinol and have unique structures, different from terrestrial plants. The natural variety of structural linkages between phloroglucinol units (PGU) and the number of hydroxyl groups present in the phlorotannins ensure a broad range of potential biological activities of these heterogeneous poly/oligomers [6,7,8]. In addition, selected independent variables such as habitat, harvesting time, light intensity exposure and nutrient availability affect the concentration of phlorotannins which are reported to constitute from 0.5 up to 20% of the biomass dry weight in different brown algae species [8]. Still, the specific types of the compounds are seldomly reported, often limiting evaluations to comparisons of total phenolic content, and rather few studies have focused on the identification of individual phlorotannins [9,10,11,12].

Numerous extraction methods have been applied with the aim to study phlorotannins from brown algae, in particular solvent extractions using a hydrophilic solvent such as water, ethanol, and methanol [13]. These are usually combined with analysis of the total phenolic content (TPC) by the Folin–Ciocalteu (F-C) assay, which is a simple and fast colorimetric method [14]. However, this method provides no information regarding the chemical composition of the phlorotannins, and is known to oxidize several non-phenolic compounds. Nuclear magnetic resonance (NMR) spectroscopic methods have sometimes been used to obtain structural information on phlorotannin structures [15,16], which is challenging since relatively large amounts of pure compound fractions need to be isolated. In this work we are instead exploring ultra-high-performance liquid chromatography/high resolution tandem mass spectrometry (UHPLC/HRMS^2^), being a method for quick profiling of phlorotannins from brown seaweeds based on their degree of polymerization. This method has thus far been used for profiling of phlorotannin-enriched extracts from five brown algae species (*Ascophyllum nodosum*, *Pelvetia canaliculata*, *Fucus spiralis*, *Fucus vesiculosus*, and *Saccharina longicruris*) [17]. Moreover, 42 phlorotannins from *Sargassum fusiforme* have been detected and identified, based on their degree of polymerisation (DP), by Li and coworkers, including fuhalol-type phlorotannins, phlorethols, fucophlorethols, and eckol-type phlorotannins [11].

The aim of this study was to analyse phlorotannins in *Saccharina latissima*, a brown seaweed which has been cultivated in North Atlantic waters with high growth rate [18]. According to previous reports, this species contains a significant level of bioactive molecules [7,19,20], but to our knowledge no study has yet identified individual phlorotannins extracted from *S. latissima.* To allow a relevant comparison, *A. nodosum* was included as an additional target, being proposed as a rich source of phlorotannins [21,22]. Solid–liquid extractions were made using both *S. latissima* and *A. nodosum* to obtain material for analysis of phlorotannin compounds by the UHPLC-HRMS^2^ method. This approach introduces identification of compounds from a species (*S. latissima*) not previously characterized concerning phlorotannin molecules, and a species (*A. nodosum*) with identifiable phlorotannins, validating the UHPLC-HRMS^2^ method used here.

## 2. Results and Discussion

### 2.1. Single Step Ethanol/Water Extraction from S. latissima

Despite being a species that is successfully cultivated in the North Atlantic, phlorotannin compounds from *Saccharina latissima* have not yet been identified. Extractions from dried milled *S. latissima* were subsequently made, and the resulting total phenolic content (TPC) was screened using the (F-C) assay to allow a fast verification of the presence of phenolic compounds (including phlorotannins) prior to further analysis. The highest yield of TPC from macroalgae has thus far been reported in extractions using either water or ethanol as solvent. Water extractions have been used for the species *Eisenia bicyclis*, *Sargassum fusiforme*, *Saccharina japonica*, and *Undaria pinnatifida*, while extractions in aqueous ethanol (30% *v*/*v*) are reported for *S. fusiforme* [11,13,23]. As our purpose was to obtain sufficient quantities of phlorotannins for identification, these two solvents and conditions were selected for the current trials. Moreover a third condition was also included: Use of aqueous ethanol (80% *v*/*v*) in a long extraction trial (20 h at 25 °C, followed by treatment of remaining solids for 10 h at 65 °C), mimicking a pretreatment used to dissolve and remove low molecular weight compounds (including aromatic compounds) prior to extraction of polysaccharides from seaweeds [24]. Use of the latter extraction methodology would allow inclusion of the phlorotannin extraction as a first step in sequential processing of seaweeds.

The three extracts were lyophilized and the presence of phenolic compounds was analysed for TPC by the (F-C) assay and by UHPLC-UV/Vis, in both cases using phloroglucinol as a standard. Despite a TPC corresponding to 3.4–7.3 mg phloroglucinol equivalents (PGE)/g extract (Figure 1), no peak corresponding to the phloroglucinol standard could be detected by UHPLC-UV/Vis in any of the samples, and the only detectable peak in the extracts was eluted very early (Appendix A). Analysis by UHPLC/HRMS^2^ was also unsuccessful, with insufficient quantities to identify any compounds (data not shown).

From this data, it can be concluded that the observed TPC using the (F-C)-assay was not corresponding to phlorotannins. The apparent phenolic content detected by the (F-C)-assay, might instead be due oxidation of non-phenolic compounds in the *S. latissima* extract (detectable as early peaks by UHPLC-UV/Vis Appendix A). The (F-C) assay is based on one- or two-electron reduction reactions between the (F-C) reagent and TPC. As the assay originally was designed for determination of total protein concentration and was later modified for determination of TPC, it is not completely specific for phlorotannins or related phenolic compounds. The (F-C) reagent in the assay may also react with e.g., protein and mannitol. *S. latissima* contains protein (up to 11% of dry material) [25] and mannitol (up to 16% of dry material) [26] which could be co-extracted with phenolic compounds and react with the (F-C) reagent, skewing the TPC result from this method [27,28].

### 2.2. Ethanol/Water Extraction from S. latissima and A. nodosum Combined with a Two-Step Purification by Solid Phase Extraction (SPE)

To obtain sufficient amounts of phlorotannins for analysis, extraction trials were subsequently repeated and combined with a two-step purification to minimize cross-reactivity in the (F-C)-assay and to obtain more concentrated samples of higher purity for analysis. The purification consisted of two solid phase extraction (SPE) steps in cartridges with C-18 silica sorbent, as described by Rajauria [29].

Dried, milled samples of both *Saccharina latissima* (in water and in ethanol: 30% and 80% *v*/*v*) and *Ascophyllum nodosum* (in ethanol: 30% and 80% *v*/*v*) were used in the extraction trials. *A. nodosum* was included at this stage, to include a species reported to be rich in phlorotannins [21,22]. Five trials were set up (three with *S. latissima* and two with *A. nodosum*) in parallel experiments at a solid:liquid ratio of 1:10 (but concerning time and temperature, adhering to the extraction procedure used in 2.1. which was 1 h and 65 °C for water extraction, 30 min and 25 °C for aqueous ethanol (30%, *v*/*v*) extraction, and (20 h at 25 °C + 10 h at 65 °C) for aqueous ethanol (80%, *v*/*v*) extraction). The 1:10 ratio was selected (and used in all subsequent trials) to balance the desire for a high solid:liquid ratio with the need to limit viscosity, as the 1:5-ratio in the trial above (Section 2.1) resulted in a very viscous extract. Higher yields of extracted phlorotannins have been reported from extractions at higher solid:liquid ratio [30], but the differences in yield were not significant within a solid:liquid ratio range from 1:5 to 1:25. A high solid:liquid ratio is, however, beneficial to avoid a long concentration process that might oxidize the phlorotannins [11].

The five resulting extracts were purified by SPE, and the eluted samples were lyophilized and analysed for TPC by the (F-C) assay, by UHPLC-UV/Vis and by UHPLC/HRMS^2^. The TPC content in the SPE eluates from the *S. latissima* extracts, increased one order of magnitude compared to the concentrations obtained directly after extraction (Figure 1), and were 83 ± 0.92, 74 ± 0.15, and 66 ± 0.15 mg PGE/g lyophilized eluate, showing only small differences between samples extracted in water, 30% *v*/*v* ethanol, and 80% *v*/*v* ethanol, respectively (Figure 2). The TPC in the *A. nodosum* eluate was one order of magnitude higher than in the corresponding *S. latissima* eluate (Figure 2), and the amount obtained was somewhat lower in extracts using the higher concentration of ethanol at long extraction time.

The UHPLC-UV/Vis chromatograms of the *S. latissima* and *A. nodosum* eluates showed several peaks, also including a peak corresponding to the previously observed first peak of potentially interfering compounds (Appendix A), although at significantly lower intensity (Appendix A). A number of other peaks were also observed, including peaks approximately corresponding to the elution time of phloroglucinol (0.9 min), and a peak with an approximate retention time of 7 min, observed in all extracts from *S. latissima* at all wavelengths (Appendix A). The chromatograms from the *A. nodosum* eluate displayed surprisingly low total peak areas (Appendix A) that did not correspond to the high TPC obtained by the (F-C) assay. The reason for this is unclear, and arises the suspicion that storage of the samples influences the stability of the targeted compounds, as the *A nodosum* samples were stored for a longer time period (at −20 °C) than the *S. latissima* samples, prior to analysis. Matching of peaks with the available chemical standards, however, was not possible using the UHPLC-UV/Vis method. It is well-known that identification of compounds in complex samples exclusively based on matching retention times with chemical standards is a quite limiting and costly method.

Hence, instead, UHPLC/HRMS^2^ was used (Section 2.3, below). Analysis of the eluates after purification of the phenolic compounds extracted from the two seaweed species using aqueous ethanol (80%, *v*/*v*) at 25 °C for 20 h, and an additional 10 h at 65 °C, was the first choice for several reasons. Firstly, although the yield of extracted phlorotannins (as interpreted from the TPC) was somewhat lower than the TPC after extraction in water or in ethanol (30%, *v*/*v*) (Figure 2), the amount of mannitol (that may interfere, with e.g., the TPC assay) was also lower in this sample according to HPAEC-PAD analysis (Appendix A). The concentration of mannitol (the only compound detected by HPAEC-PAD) was 1.1 mg/mL in the eluate originating from the aqueous ethanol (80%, *v*/*v*) extract, while the samples extracted by aqueous ethanol (30%, *v*/*v*) or water contained 2.3 mg mannitol/mL eluate in both cases. Secondly, (as stated above) these extraction conditions can be used as a first step in a seaweed cascading process, prior to extraction of polysaccharides, simplifying refining of the seaweed biomass.

### 2.3. Identification of Phlorotannins by UHPLC/HRMS^2^

Five phlorotannins were identified, four in *A. nodosum* and one in *S. latissima*, and their identification confidence levels according to Schymanski et al. [31] are shown in Table 1. The base peak ion-chromatograms, monitoring the most intense peaks in each mass spectrum, are shown for both seaweed species in Figure 3. The data revealed a previously not identified phlorotannin from *S. latissima*, and the extracted ion chromatograms and comparisons of detected and theoretical isotope patterns for this phlorotannin areshown in Figure 4. The *S. latissima* phlorotannin had a lower molecular mass (Table 1), than the four identified compounds from *A. nodosum*, for which the corresponding chromatograms and patterns are shown in the Appendix A. For all five identified compounds a good match of the detected and theoretical isotope pattern was observed.

The extracted ion-chromatogram of the tentative phlorotannin identified in *S. latissima* with *m*/*z* 265.0352 in Figure 4A shows a peak with retention of 6.43 min. The comparison of the detected and theoretical isotope pattern of *m*/*z* 265.0352 shown in Figure 4B shows an overlap of the detected ^13^C-peak at *m*/*z* 266.0387 with a mass difference of 1.1 mDa to the theoretical *m*/*z*. The MS^2^ spectra were obtained for three identified compounds in *A. nodosum* showing typical fragmentation patterns for phlorotannins (Figure 5). In the MS^2^ spectrum, three fragments were identified. At *m*/*z* 315.02 a neutral loss of C_2_H_2_O_3_ was observed, which most likely corresponds to a loss of two carbon monoxide and one water molecule. A neutral loss of C_6_H_5_O_3_ corresponding to a loss of a phloroglucinol unit (PGU) was observed at *m*/*z* 264.02. The third fragment observed at *m*/*z* 249.02 showed a neutral loss of C_6_H_4_O_4_, which corresponds to a PGU with one additional oxygen atom (Figure 5). Similar fragmentation patterns were obtained for *m*/*z* 497.0719 and 621.0875 and the MS^1^ and MS^2^ spectra (Appendix A).

Tentative chemical structures of all identified phlorotannins are illustrated in Figure 6. In *A. nodosum* the four identified phlorotannins (Table 1) were in agreement with the structures reported by Tierney [17]. The *A. nodosum* structures identified here were, however, built from 3 to 6 PGU, in contrast to the observation in the profiling by [17] that this species mainly contained phlorotannins built from 6 to 13 PGU with very low amounts of phlorotannins of lower degrees of polymerization (DPs). This observation indicates that factors like habitat, harvesting time, light intensity exposure and nutrient availability may not only affect the concentration of phlorotannins, as reported by [8], but also the composition of phlorotannins. Another explanation is that there might be a bias against larger molecular weight phlorotannins during the extraction, chromatography or electrospray ionisation in MS.

Phlorotannin molecules with a *m*/*z* 621.0875 corresponding to 5 PGU has previously been reported from *Sargassum fusiforme.* In addition, Catarino et al. and Hermund et al. have reported the presence of compounds with *m*/*z* 497.0719, 621.0875 and 745.1026 (corresponding to 4–6 PGU) in the algae *Fucus vesiculosus* [12,21], showing that phlorotannins with *m*/*z* corresponding to the molecules found in *A. nodosum* were also present in other species.

In *S. latissima* one tentative phlorotannin with *m*/*z* 265.0352, was identified with the identification confidence level 4 (Table 1). The elemental composition of this compound corresponds to the compound bifuhalol, which is a small molecule built from two PGU. No phlorotannin structure corresponding to that currently identified has previously been reported from this species of macroalgae. Structures built from two PGU (with *m*/*z* 265.0352 as in the *S. latissima* extract investigated here) and three PGU (*m*/*z* 389.0509 as in the *A. nodosum* extract) have, however, been reported from the brown algae *S. fusiforme* [11]. This again shows that the different types of phlorotannin structures are conserved in the different types of brown macroalgae, although at different concentrations and with different profiles in the different species.

## 3. Materials and Methods

### 3.1. Materials

A batch of seaweed powder (S-B-1707-D-1-170914-5-L) as fine ground seaweed biomass of *Saccharina latissima* was obtained from Ocean rainforest (Kaldbak, Faroe Islands). Dried and milled *Ascophyllum nodosum* was obtained from Thorsverk, Iceland.

All chemicals were supplied from Sigma-Aldrich (Stockholm, Sweden) unless otherwise stated.

### 3.2. Solvent Extraction

Phlorotannin extraction trials from *S. latissima*, were made using the dried milled *S. latissima* and were set up at three extraction conditions: (i) Using water at 70 and 65 °C in 1 h extraction trials at a solid:liquid ratio of 1:100 and 1:10, respectively, (ii) using aqueous ethanol (30% *v*/*v*) at 25 °C in 30 min trials at a solid:liquid ratio of 1:5 and 1:10, respectively, and (iii) using ethanol (80% *v*/*v*) in a long extraction trial (20 h at 25 °C + 10 h at 65 °C) with the solid ratio of 1:10.

The same procedure, also, was applied for *A. nodosum* using ethanol (30%, *v*/*v*) and (80%, *v*/*v*), in separate trials, with the solid:liquid ratio of 1:10.

#### 3.2.1. Extraction of Phlorotannins by Water

Extraction of phlorotannins by water was performed using different solid:liquid ratios and temperatures. The first experiment was using a solid:liquid ratio of (1:100) [23]. Ten millilitres of distilled water were added to 0.1 g of seaweed in a 20-mL glass bottle with screw cap. Then, the bottle was covered with foil and put in a heating block at 70 °C for one hour with constant shaking at 700 rpm. A second experiment was carried out at a solid:liquid ratio of (1:10). Fifty millilitres of distilled water were added to 5 g seaweed in a 100-mL glass bottle with screw cap. Then the bottle was covered with foil and put in a shaker incubator (Ecotron, Infors, Bottmingen-Basel, Switzerland) at 65 °C for one hour with constant shaking at 150 rpm. After that the samples from both experiments were centrifuged (3890× *g*, 10 min, 4 °C) to separate the supernatants from the seaweed residue. The supernatants were freeze-dried and kept in a freezer at (−20 °C) at dark conditions for further analysis and experiments. The experiments were performed in triplicates.

#### 3.2.2. Extraction of Phlorotannins by Ethanol

Extraction of phlorotannins by ethanol was performed using ethanol two different concentrations (30% and 80% *v*/*v*), two temperatures (25 and 25 + 65 °C), and times (1 h or 20 + 10 h).

The experiments with ethanol (30%, *v*/*v*) were carried out with a solid:liquid ratio of (1:5, 40 g:200 mL) [11] and (1:10, 5 g:50 mL) in two separate glass bottles with screw cap. After introducing the seaweed, the bottles were covered with foil and put in the shaker incubator at 25 °C for 30 min with constant shaking at 150 rpm. After that, the samples were centrifuged (3890× *g*, 10 min, 4 °C) and the supernatant was separated from the seaweed residue. Then, the ethanol was removed from supernatants by a rotary evaporator (Heidolph, Schwabach, Germany) and the supernatants were freeze-dried using a lyophilizer (Labconco, Kansas City, MO, USA) and kept frozen (−20 °C) in dark conditions for further analysis and experiments.

The experiment with ethanol (80%, *v*/*v*) was carried out with a solid:liquid ratio of (1:10). Fifty millilitres of ethanol (80%, *v*/*v*) was added to 5 g seaweed in a 100-mL glass bottle with screw cap. Then the bottle was covered with foil and put in the shaker incubator at 25 °C for 20 h and then at 65 °C for 5 h with constant shaking at 150 rpm. Then the sample was centrifuged (3890× *g*, 10 min, 25 °C) and the supernatant was separated from the solid part and collected. Another 50 mL of ethanol (80%, *v*/*v*) was added to the solid part and the bottle was covered with foil and put in the shaker incubator at 65 °C for extra 5 h with constant shaking at 150 rpm. After that the sample was centrifuged at (3890× *g*, 10 min) and 4 °C and the supernatants from both steps were mixed and then, after removal of ethanol by the rotary evaporator, the supernatant was freeze-dried using a lyophilizer and kept in a freezer (−20 °C) in dark condition for further analysis and experiments.

The experiments were performed in triplicates.

### 3.3. Two-Step Purification of Phlorotannins by Solid Phase Extraction (SPE)

The extracted liquid samples from solvent extractions were filtered through 0.2 μm polypropylene syringe filters before the purification step. A two-step purification was performed using two SPE cartridges and the elution solutions used for eluting of the polar and non-polar compounds were the same as the mobile phases used for UHPLC-U V/Vis analysis (below).

The first step of purification was done using the Sep-pak-C18 SPE end capped cartridge (10 g sorbent mass, 55–105 μm, Waters), which was initially conditioned with a full bed volume of 100% acetonitrile and then with 0.25% *v*/*v* aqueous acetic acid. The filtered aqueous sample was loaded onto the column and allowed to flow through under gravity. After that the co-extracted compounds such as mannitol and other polar compounds were eluted by 100 mL of 0.25% *v*/*v* aqueous acetic acid. Then the phenolic compounds were eluted using approximately 200 mL of acetonitrile:water (80:20%, *v*/*v*) containing 0.25% *v*/*v* acetic acid. After that, the acetonitrile in the collected fractions was removed by rotary evaporator and the remaining liquid fraction was filtered through 0.2 μm polypropylene syringe filters before the second purification step.

The second purification step was carried out using DSC-18 SPE cartridge (Supelco, Sigma-Aldrich, Sweden) with the same protocol as above. After removing acetonitrile from the collected fraction of second purification step, the fraction was freeze-dried using a lyophilizer and kept in a freezer at (−20 °C) in dark conditions for further analysis.

### 3.4. Analytical Procedures

#### 3.4.1. Determination of Total Phenolic Content (TPC)

Total phenolic content of the seaweed extracts was determined by the Folin–Ciocalteu (F-C) assay described by Wang with modifications [11,32]. The TPC analysis was done from lyophilized extracted supernatant or directly from the liquid (after filtration using 0.2 μm syringe filters). An aqueous solution with a concentration of 1 g/L of the lyophilized extract was prepared and filtered through 0.2 μm syringe filters. Then 0.2 mL of the prepared solution from the lyophilized extract or the liquid supernatant was added to 1.3 mL of distilled water and 0.5 mL of Folin–Ciocalteu’s phenol reagent. After mixing, 1 mL of 7.5% (*w*/*v*) Na_2_CO_3_ was added to the solution. The solution was mixed on a vortex mixer and incubated for 1 h at room temperature in darkness. Then, the absorbance of the sample was measured at 770 nm using a UV/Vis spectrophotometer (Ultrospec 1000, Biochrom Ltd., Cambridge, UK). The compound phloroglucinol was used as standard and a standard curve was made with serial phloroglucinol dilutions (5–100 μg/mL). The result was obtained as milligrams of phloroglucinol equivalents (PGEs) per gram extract for lyophilized samples.

#### 3.4.2. Qualitative Analysis of Phlorotannins by UHPLC-UV/Vis

Qualitative analysis of phlorotannins was performed using a UHPLC-UV/Vis system (Ultimate-3000 RSLC, Dionex, ThermoFisher Scientific, Waltham, MA, USA) equipped with an online degasser, a duel gradient solvent pump, a thermostatic controlled autosampler and column oven, and a UV-Vis detector. The system was connected to a C18 column (Gemini-NX, 100 × 2 mm, 3 μm particle size, 110Å, Phenomenex, Værløse, Denmark). Optimization of the UHPLC analysis has been described by Rajauria [29]. The mobile phase consisted of (A) acetonitrile:water (80:20%, *v*/*v*) and (B) ultrapure water, both containing 0.25% *v*/*v* acetic acid. The gradient started with 10% (A) and was kept for 20 min and then increased to 20% for 10 min and then 30% for 10 min, and then 0% for 10 min. The flow rate was 0.5 mL/min and the column oven maintained at 25 °C. Separation was performed by injection of 20 μL of the sample. Eluted phlorotannins were detected at 254 nm, 280 nm, and 320 nm, using a UV/Vis detector (Ultimate 3000 RS, Dionex, ThermoFisher Scientific, Waltham, MA, USA). The aqueous solution of the lyophilized extract (1 g/L) was prepared, filtered using 0.2 μm syringe filters in a HPLC vial and used for analysis. Phloroglucinol was used as standard with the concentrations 50, 100, 200, and 500 mg/L.

#### 3.4.3. Sugar Analysis

Sugar analysis of the liquid extracts of *S. latissima* was performed using a high-performance anion exchange chromatography system equipped with pulsed amperometric detector (HPAEC-PAD). A Dionex CarboPac PA-20 analytical column (150 mm × 3 mm, 6 µm) and guard column (30 mm × 3 mm) (Thermo Fisher Scientific, Waltham, MA, USA) were used to separate sugars. Eluents used were: (A) Ultrapure water; (B) 2 mM sodium hydroxide and (C) 200 mM sodium hydroxide. Separation of monomeric sugars and sugar alcohol mannitol was performed under isocratic conditions at a flow rate of 0.5 mL/min for 30 min using an eluent mixture of 62.5% A, 37.5% B. The sample was centrifuged (3890× *g*, 10 min, 4 °C) and was, after proper dilution, filtered through 0.2 µm polypropylene syringe filter into a plastic vial and analysed.

#### 3.4.4. Analysis of Phlorotannins by UHPLC/MS^2^

All samples were injected into a Thermo Scientific Accela UHPLC system equipped with an Accela Autosampler, an Accela 600 pump, and an Accela PDA detector (Thermo Scientific, Waltham, MA, USA). The UHPLC system was connected to an LTQ Orbitrap Velos Pro mass spectrometer equipped with a heated electrospray ionisation source (Thermo Scientific). An ACQUITY CSH Phenyl-hexyl UPLC column (2.1 mm × 100 mm, 1.7 µm, 130 Å) protected with an ACQUITY UPLC VanGuard pre-column (CSH Phenyl-hexyl (2.1 mm × 5 mm, 1.7 µm, 130 Å) were purchased from Waters (Milford, MA, USA). For centrifugation of the samples a 5424R Eppendorf centrifuge (Eppendorf, Hamburg, Germany) was used.

Xcalibur 2.2 (Thermo Fisher Scientific) was used to operate the UHPLC/MS system and for data acquisition. The open source software MZmine 2 and Xcalibur 2.2 were used for MS data evaluation. The lyophilized samples were dissolved in water/acetonitrile (95/5, *v*/*v*) with a final concentration of 10 mg/mL. Then the samples were centrifuged for 10 min at 14.000 rpm and 20 °C and then, five microliters of each sample were injected. After each injection the syringe, injection needle and the injection transfer tubes were flushed with 1 mL with an acetonitrile/water (50/50, *v*/*v*) flush solution. As mobile phase solvent (A) water and (B) acetonitrile/water (95/5, *v*/*v*), both containing 10 mM ammonium formate were used. A gradient elution was performed starting with 5% B at 0 min, and was then hold up to 15 min, then ramped up to 30% B until 30 min, then ramped up to 100% B until 35 min, then hold up 47 min. Then the mobile phase composition was ramped down to 5% B until 48 min and hold until 58 min to equilibrate the column for the next injection. The column temperature was 40 °C and a flow rate of 0.3 mL/min was used. UV/Vis spectra were collected at 210 nm, 254 nm and 600 nm, each with a filter bandwidth of 9 nm and a sample rate of 20 Hz.

Spectral data were collected with a photo diode array (PDA) detector from 200 nm to 600 nm using a wavelength step of 1 nm, a sample rate of 20 Hz and a filter bandwidth of 9 nm.

For the ionisation a heated electrospray ionisation source was used in negative mode, using a spray voltage of 3 kV, a heater temperature of 275 °C, a sheath gas flow of 70 (arbitrary units), an auxiliary gas flow of 10 (arbitrary units), a sweep gas flow rate of 0 (arbitrary units), a capillary temperature of 275 °C, and a source fragmentation voltage of 0 V.

The mass spectrometer was used in data-dependent MS^2^ mode, performing a MS^2^ for the three most abundant ions in every scan event. Each MS^1^ scan event was performed in the Orbitrap mass analyzer with a MS resolution of 100.000 and every MS^2^ scan event was performed using the linear quadrupole ion trap. The *m*/*z* scan range was set to *m*/*z* 120 to 1500. Collision induced dissociation was performed using a default charge state of 2, activation time of 10 ms, an isolation with of *m*/*z* 2.0, an activation q of 0.250 (arbitrary units) and a normalized collision energy of 35%.

Before every sample batch the mass spectrometer was calibrated using an external calibration standard. To create sample peak lists and to determine chemical formulas, including C, H and O, and the ring double bond (RDB) equivalents MZmine 2 was used. A detailed overview of the used MZmine 2 workflow is shown in (Appendix A).

#### 3.4.5. Identification of Phlorotannins from the UHPLC/MS^2^ Analysis

The identification confidence level system introduced by Schymanski et al. [31] was used to categorize the identification confidence of identified phlorotannins. The system by Schymanski et al. consists in brief of five identification confidence levels, where level 1 represents the highest identification confidence and level 5 the lowest.

Level 1 is given if an identified compound can be compared with a reference standard with respect to retention time, *m*/*z* and MS^n^ fragmentation.

Level 2 is given, if for an identified compound an exact mass, an unequivocal molecular formula, a matching isotope pattern is obtained, and an MS^n^ fragmentation pattern is matching with an MS^n^ fragmentation reported in literature.

At level 3 an exact mass, an unequivocal molecular formula, a matching isotope pattern and a compound specific MS^n^ fragmentation pattern are needed.

For level 4 an exact mass, an unequivocal molecular formula, and a matching isotope pattern needs to be obtained.

At level 5 only the exact mass is known.

## 4. Conclusions

In this study, extraction and identification of phlorotannins from *S. latissima* and *A. nodosum* were made. Different extraction methods were used and among them, phlorotannin extraction using ethanol (80%, *v*/*v*) solution was prioritized, as it allows separation of phenolic compounds, prior to extracting the macroalgeal polysaccharides in a biorefinery sequence.

Purification of the extracts was needed to concentrate the extracted phlorotannins, and to remove co-extracted compounds (mainly mannitol). Successful identification of compounds from the purified extract was accomplished using UHPLC/MS^2^, which verified the presence of four phlorotannins in *A. nodosum* with the formulas C_18_H_14_O_10_ (bifuhalol-type), C_24_H_18_O_12_ (phlorethol-type), C_30_H_22_O_15_ (fucophlorethol-type), and C_36_H_26_O_18_ (phlorethol-type). One phlorotannin in *S. latissima* was verified with the formula C_12_H_10_O_7_ which fits with the molecular weight and elemental composition of bifuhalol.

## Figures and Tables

**Figure 1 molecules-26-00043-f001:**
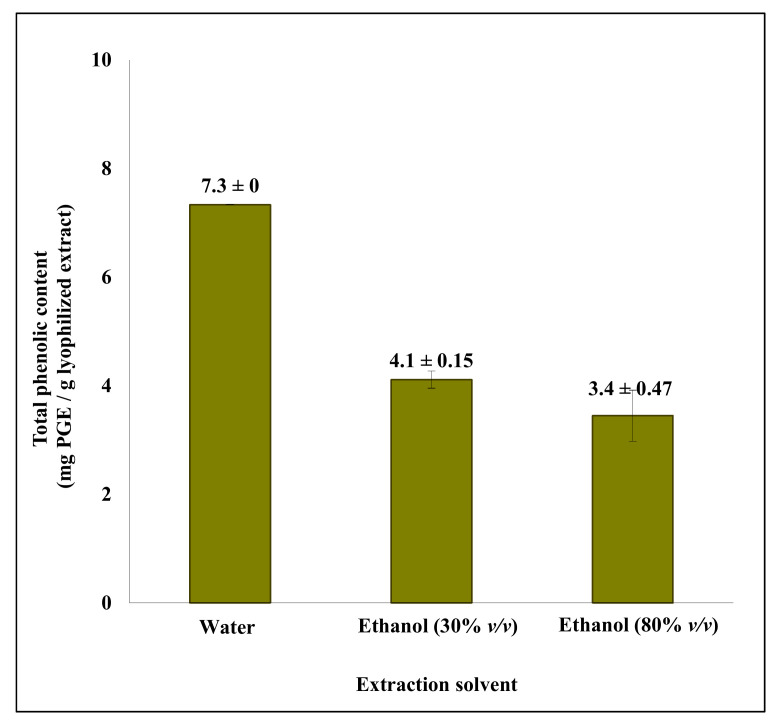
Total phenolic content (TPC) of lyophilized crude extracts (1 g/L) from solvent extraction of *S. latissima*. Solid:liquid ratio was (1:100) for extraction by water, (1:5) for extraction by aqueous ethanol (30% *v*/*v*), and (1:10) for extraction by aqueous ethanol (80% *v*/*v*). The results were obtained as milligrams of phloroglucinol equivalents (PGEs) per gram extract. Data are mean ± SD corresponding to two experimental replicates.

**Figure 2 molecules-26-00043-f002:**
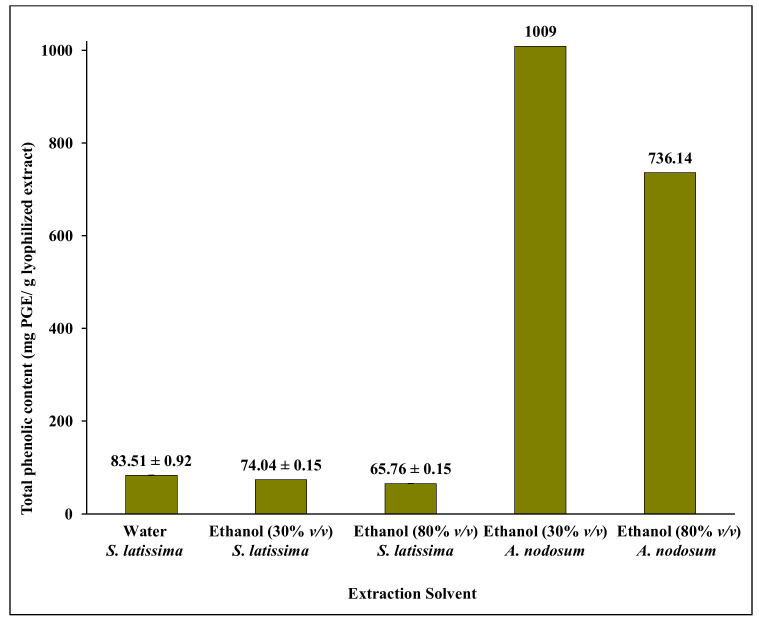
Total phenolic content (TPC) of lyophilized purified extracts (1 g/L) from solvent extraction of *S. latissima* and *A. nodosum* after two step purifications. Data for *S. latissima* are mean ± SD corresponding to two experimental replicates and data for *A. nodosum* are corresponding to single experiments.

**Figure 3 molecules-26-00043-f003:**
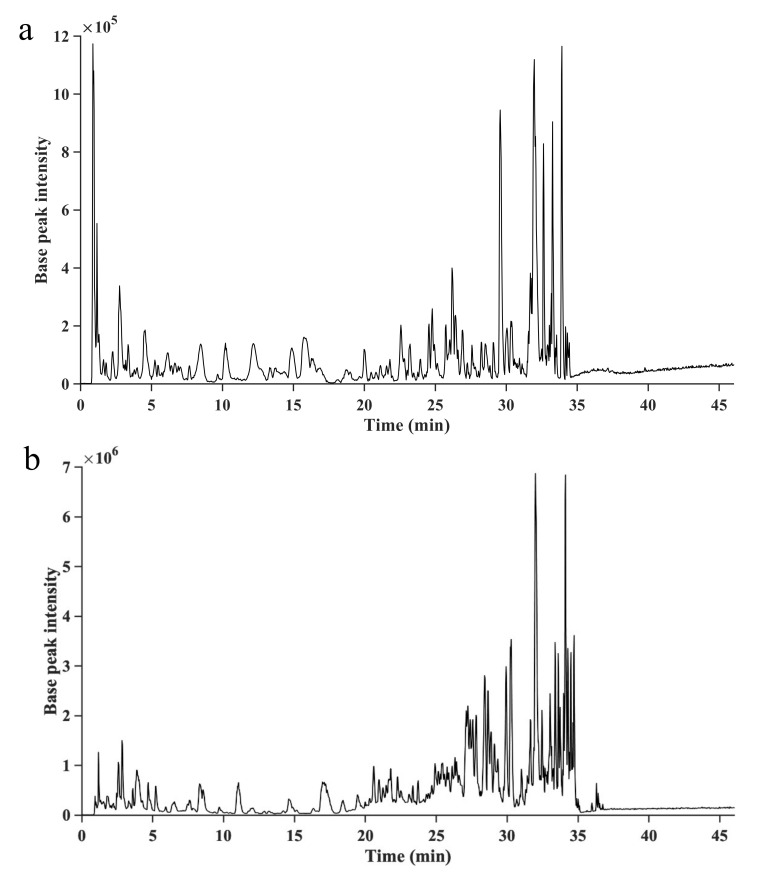
Full scan base peak ion-chromatograms of *A. nodosum* (**a**) and *S. latissima* (**b**) extracted with aqueous ethanol (80%, *v*/*v*).

**Figure 4 molecules-26-00043-f004:**
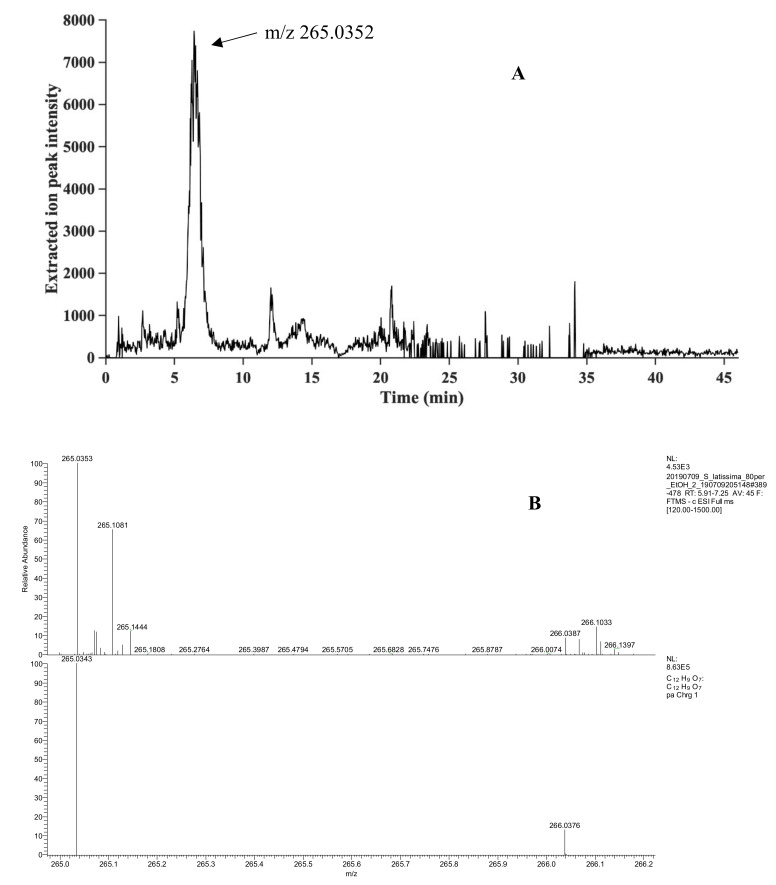
(**A**) Extracted ion-chromatogram of *m*/*z* 265.0352 found in *S. latissima*. (**B**) Comparison of detected (**top**) and calculated (**bottom**) isotopes for *m*/*z* 265.0352 found in *S. latissima.*

**Figure 5 molecules-26-00043-f005:**
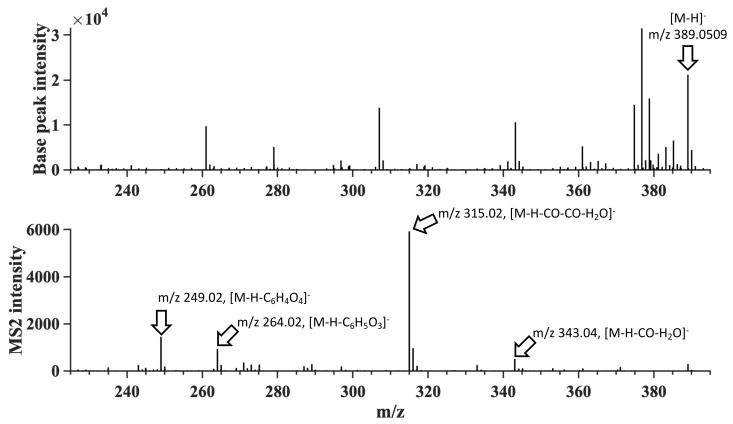
MS^1^ and MS^2^ spectra of *m*/*z* 389.0509 found in *A. nodosum*.

**Figure 6 molecules-26-00043-f006:**
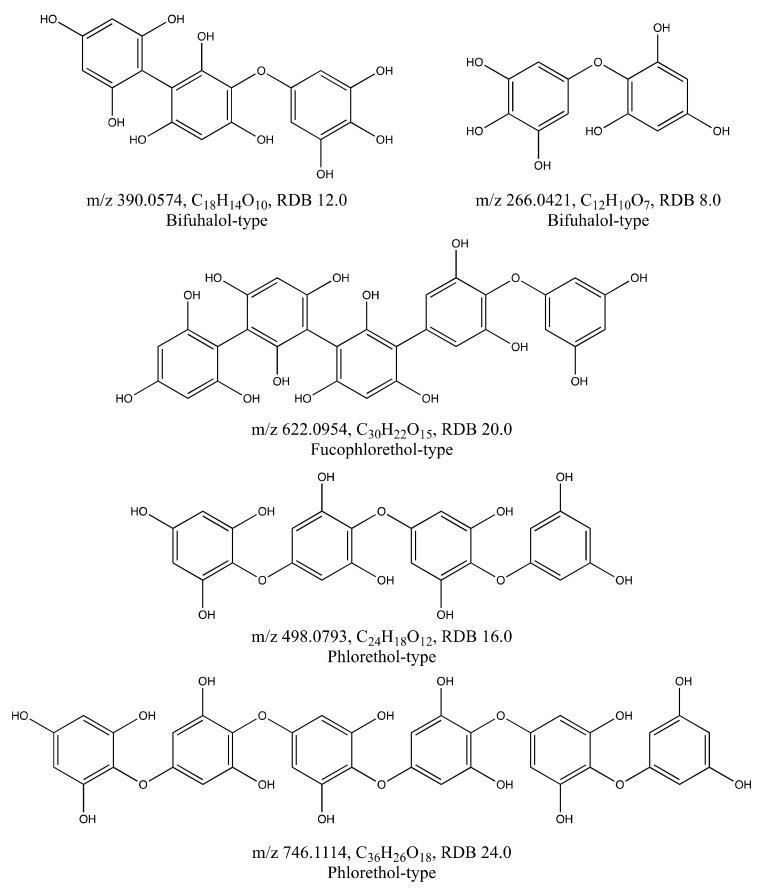
Tentative chemical structures of phlorotannins found in *A. nodosum* (*m*/*z* 390.0574, 622.0954, 498.0793 and 746.1114) and *S. latissima* (*m*/*z* 266.0421). RDB = ring double bond equivalent.

**Table 1 molecules-26-00043-t001:** Tentatively identified phlorotannins in *A. nodosum* and *S. latissima* in samples extracted with aqueous ethanol (80%, *v*/*v*) with detected *m*/*z*, theoretical *m*/*z*, difference in *m*/*z*, retention time (RT), determined chemical formula, ring double bond (RDB) equivalent for neutral compounds, detected MS^2^ fragments and identification level according to Schymanski et al. [31]

Species	Detected [M − H]^−^	Theoretical [M − H]^−^	Mass Difference (mDa)	RT (min)	Chemical Formula (Neutral)	RDB	Detected MS^2^ Fragments and Proposed Neutral Losses	Identification Confidence Level
*A. nodosum*	389.0509	389.0501	−0.8	21.03	C_18_H_14_O_10_	12.0	343.04 (−CO, −H_2_O); 315.02 (−2 × CO, −H_2_O); 264.02 (−C_6_H_5_O_3_); 249.02 (−C_6_H_4_O_4_)	2
	497.0719	497.0720	0.1	9.72	C_24_H_18_O_12_	16.0	479.20 (−H_2_O); 413.02 (−3 × CO); 373.07 (−C_6_H_4_O_3_); 371.03 (−C_6_H_6_O_3_); 353.01 (−C_6_H_6_O_3_, −H_2_O); 265.01 (−C_12_H_8_O_5_); 229.13 (−C_12_H_8_O_5_, −2 × H_2_O)	2
	621.0875	621.0881	−0.6	19.56	C_30_H_22_O_15_	20.0	495.10 (−C_6_H_6_O_3_); 371.04 (−C_12_H_10_O_6_)	2 *
	745.1026	745.1041	1.5	12.17	C_36_H_26_O_18_	24.0		4
*S. latissima*	265.0352	265.0348	−0.4	6.43	C_12_H_10_O_7_	8.0		4

* Obtained MS^2^ data from ion-trap MS experiment not clear due to overlapping MS peaks (see SI).

## Data Availability

The data presented in this study are available on request from the corresponding author after approval by relevant partners in the Macro Cascade project.

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
