# Peer review of "Identification of Phlorotannins in the Brown Algae, Saccharina latissima and Ascophyllum nodosum by Ultra-High-Performance Liquid Chromatography Coupled to High-Resolution Tandem Mass Spectrometry"

_molecules, 2020, doi:10.3390/molecules26010043_

Round 1

Reviewer 1 Report

The author has improved the manuscript thus the actual version is adequate for a publication in Molecules

Author Response

Answer: Thanks a lot for reviewing our manuscript

Reviewer 2 Report

General Comments

I suggest Authors to add the identification of peaks in Figure 3 e and Fugure 4. Moreover Authors must add in the text of paper UHPLC/HRMS chromatogram of standars for identified compounds and for standards analysed by HPAEC-PAD. In supplementary materials too peaks are not identified. However, behind MS identification chromatograms of standars of major compounds has to be reported to asses the right identification.

For this reason I suggest to accept the paper after major revisions.

Author Response

Responses to the Reviewers´ Comments:

Title: Identification of phlorotannins in the brown algae, Saccharina latissima and Ascophyllum nodosum by ultra-high-performance liquid chromatography coupled to high-resolution tandem mass spectrometry

Comments and Suggestions for Authors

General Comments

I suggest Authors to add the identification of peaks in Figure 3 e and Fugure 4. Moreover Authors must add in the text of paper UHPLC/HRMS chromatogram of standars for identified compounds and for standards analysed by HPAEC-PAD. In supplementary materials too peaks are not identified. However, behind MS identification chromatograms of standars of major compounds has to be reported to asses the right identification.

For this reason I suggest to accept the paper after major revisions.

Answer: We acknowledge the valuable comments from the reviewer. In Figure 4 and the corresponding Figures in the supplementary materials we added peak identities to the extracted ion-chromatograms for a clearer identification as suggested by the reviewer.

In Figure 3, we think that due to the high sample complexity and the relative low intensities of the identified phlorotannins in the extracted ion-chromatograms, peak labelling in the fullscan base peak ion-chromatograms might be misleading due to higher peak intensities of the corresponding peaks at the retention times of the identified phlorotannins most likely due to co-elution of other components of the samples.

We agree with the reviewer that a confirmation of an identified compound in a sample with a reference standard gives the highest identification confidence. Unfortunately, for the identified phlorotannins in the S. latissima and A. nodosum samples no reference standards are commercially available. Therefore, we applied the identification confidence level system suggested by Schymanski et al (Environ. Sci. Technol. 2014, 48, 2097-2098) to present the reader the highest transparency about the identification confidence of the identified phlorotannins in our samples.

The chromatographic profiles of mannitol as standard at different concentration of 1, 10, 30, and 50 mg/L using HPAEC-PAD system was added to the Figure S5 (Supplementary materials).

Round 2

Reviewer 2 Report

Review of Manuscript ID: molecules-1046592     Title Identification of phlorotannins in the brown algae, Saccharina latissima and Ascophyllum nodosum by ultra-high-performance liquid chromatography coupled to high-resolution tandem mass spectrometry   Authors have notably improved manuscript and so I suggesto to publish it in this form.

This manuscript is a resubmission of an earlier submission. The following is a list of the peer review reports and author responses from that submission.

Round 1

Reviewer 1 Report

The manuscript presented the study of phenolic compounds identification in brown macroalgae. Some improvements are suggested to the authors to increase the quality of the manuscript. Not only the content, but English grammar and sentence construction are also needed considerable help to improve.

Introduction:
- The storyline is still confusing—no redlining between some sentences nor some paragraphs. The information provided by the author jumps from one description into another.
- irrelevant information was also included in the background such as the mechanism of phenolics in protecting the algae during their life cycles and the following sentences. No further studies were later reported in the presented paper that dealt with this information.
- The important justifications in the need to identify phenolic compounds in Saccharina latissima and Ascophyllum nodosum are omitted as well as the studied independent variables.

Result and Discussion:
- 2.1 first paragraph: such information was briefly mentioned in the M&M section. Therefore, explaining it again in this section is unnecessary, also it should be explained in the M&M section, not R&D.
Also, apart from different solvent usage for the extraction procedure, extraction time was differentiated. However, no discussion regarding this was found in the manuscript. Then, what is the purpose of this treatment?
- Figure 1: why the author only presents the data for S. latissima while in the aim of the study proposed two species (A. nodosum)?
- Line 105: before defining the reason for the apparent data, it should be explained beforehand why such data is undesirable.
- Figure 2: why the treatment for both samples was different? The data showed that A. nodosum was only treated with 2 types of solvent, while S. latissima underwent 3 experiments using 3 different solvents.

Conclusion:
- line 400: what is the compound name did these formulas stand for?

Reviewer 2 Report

Review of manuscript: molecules-972182

Title: “Identification of polyphenols in the brown algae, Saccharina latissima and Ascophyllum nodosum”

General comments:

This manuscript is very poorly written and is very confusing both in descriptive section and in the experimentals. Furthermore, in the Supplementary Materials many GC-chromatograms are reported that do not have a good separation efficiency, as the authors themselves point out in the text.
Throughout the text the references are not correctly numbered: there is no increasing number but random references are cited, without criteria.
It is not clear whether the aim of the work is the analysis of polyphenols extracted in brown algae (Title) or the identification of particular classes of polyphenols, the phlorotannins (Text).
The solid-liquid extraction chosen by the authors is very weak: maceration and infusion of algae in water and a mixture of water and alcohol and no optimization of these preocesses is reported; in addition there is no comparison with an official extraction method such as Soxhlet.The identification of a new compound must be assessed with other instrumental techniques like NMR, when it is not possible to use standard certified compounds.
In my opinion this work must be rewritten in a better way and the compound separations must be increased in efficiency by avoiding the elution of the analytes in times corresponding to the dead volume of the chromatography column.